# Postoperative Physical Therapy Program Focused on Low Back Pain Can Improve Treatment Satisfaction after Minimally Invasive Lumbar Decompression

**DOI:** 10.3390/jcm11195566

**Published:** 2022-09-22

**Authors:** Hidetomi Terai, Koji Tamai, Kunikazu Kaneda, Toshimitsu Omine, Hiroshi Katsuda, Nagakazu Shimada, Yuto Kobayashi, Hiroaki Nakamura

**Affiliations:** 1Department of Orthopaedic Surgery, Osaka Metropolitan University Graduate School of Medicine, Osaka 545-8585, Japan; 2Department of Orthopaedic Surgery, Shimada Hospital, Osaka 583-0875, Japan; 3Graduate School of Comprehensive Rehabilitation, Osaka Prefecture University, Osaka 583-8555, Japan; 4Division of Physical Therapy, Department of Rehabilitation Sciences, Faculty of Allied Health Sciences, Kansai University of Welfare Sciences, Osaka 582-0026, Japan

**Keywords:** patient satisfaction, minimally invasive surgical procedures, decompression, physical therapy, low back pain

## Abstract

Patient satisfaction is crucial in pay-for-performance initiatives. To achieve further improvement in satisfaction, modifiable factors should be identified according to the surgery type. Using a prospective cohort, we compared the overall treatment satisfaction after microendoscopic lumbar decompression between patients treated postoperatively with a conventional physical therapy (PT) program (control; n = 100) and those treated with a PT program focused on low back pain (LBP) improvement (test; n = 100). Both programs included 40 min outpatient sessions, once per week for 3 months postoperatively. Adequate compliance was achieved in 92 and 84 patients in the control and test cohorts, respectively. There were no significant differences in background factors; however, the patient-reported pain score at 3 months postoperatively was significantly better, and treatment satisfaction was significantly higher in the test than in the control cohort (−0.02 ± 0.02 vs. −0.03 ± 0.03, *p* = 0.029; 70.2% vs. 55.4%, *p* = 0.045, respectively). In the multivariate logistic regression analysis, patients treated with the LBP program tended to be more satisfied than those treated with the conventional program, independent of age, sex, and diagnosis (adjusted odds ratio = 2.34, *p* = 0.012). Postoperative management with the LBP program could reduce pain more effectively and aid spine surgeons in achieving higher overall satisfaction after minimally invasive lumbar decompression, without additional pharmacological therapy.

## 1. Introduction

In the past several decades, patient satisfaction ratings have played an important role in reimbursement management, as part of the ongoing pay-for-performance initiative, which has resulted in the medical community placing a greater emphasis on patient satisfaction [1,2]. Furthermore, healthcare systems are often judged and ranked based on patient satisfaction metrics, with an implied association with the quality of care [3,4]. Hence, it is critical to understand the factors that may influence patient satisfaction scores.

Many factors have been reported as being related to patient satisfaction for the treatment of spinal disease; for example, the physician–patient relationship [5], sex/ethnicity concordance between the patient and physician [6], age [7], length of hospital stay [8], postoperative pain [9], preoperative opioid use [10], and psychological factors [4,11]. In addition, satisfaction may be affected by the type of surgery itself; for example, patients with spinal disease tend to be less satisfied than those with cranial disease [12], and cervical surgery may provide higher treatment satisfaction than lumbar surgery [7]. Therefore, to achieve further improvement in satisfaction ratings, modifiable factors should be identified according to each surgery type.

Several minimally invasive decompression techniques, including microendoscopic surgery, enable the preservation of anatomical structures, such as the back muscles [13]. As a result, patients treated with these techniques have reduced postoperative low back pain (LBP) compared to patients treated with standard open surgery [14]. However, it is well-known that patients still have some LBP even after such minimally invasive techniques [15,16]. Furthermore, the severity of LBP has been reported as significantly related to improvement in the mental-related quality of life (QOL), independently of the severity of the symptoms, in patients treated with lumber microendoscopic decompression [17]. Based on these studies, we hypothesized that a postoperative physical therapy program focused on LBP may improve not only the postoperative LBP, but also the overall satisfaction after minimally invasive lumbar decompression. Hence, the aim of the current prospective study was to compare the overall treatment satisfaction after microendoscopic lumbar decompression between patients treated with a postoperative physical therapy program focused on the LBP and those treated with a conventional physical therapy program.

## 2. Materials and Methods

### 2.1. Study Design

A prospective, non-randomized clinical trial was conducted (Figure 1). Initially, 100 consecutive patients undergoing microendoscopic decompression for lumbar disc herniation (LDH) or lumbar spinal stenosis (LSS) between 01 Feb 2020 and 22 Apr 2020were enrolled into the control cohort. They were treated with a conventional physical therapy program for three months postoperatively by therapists who were blinded to the study aims. Subsequently, all therapists completed several sessions to learn a new physical therapy program (i.e., the LBP program), which focused on improving LBP. A 4-week trial period to standardize the knowledge and skills of five enrolled therapists was completed in August 2020. Subsequently, 100 consecutive patients who underwent surgery under the same indications as the control cohort between 01 September 2020 and 12 November 2020 were enrolled as the test cohort. They were treated with the LBP program for 3 months postoperatively. With the exception of the physical therapy program, all patients in both groups were treated with the same postoperative protocol.

### 2.2. Perioperative Clinical Course

#### 2.2.1. Surgical Criteria

Posterior endoscopic decompression was indicated in patients with neurogenic claudication or radicular pain with associated neurological signs, stenosis, or herniation on magnetic resonance imaging at a level that explained their symptoms, and no improvement despite adequate conservative treatment for at least 3 months. The exclusion criteria were as follows: spondylolisthesis of more than grade 2, spondylolytic and spondylolisthesis, and degenerative lumbar scoliosis with a Cobb angle >20°. Surgery was performed under general anesthesia. For patients with LDH, microendoscopic discectomy was performed. Microendoscopic bilateral decompression via the unilateral approach with partial laminotomy was applied in patients with LSS [18].

#### 2.2.2. Postoperative Care

All patients were treated according to the standardized care pathways of our institution. All patients were allowed to sit and walk at 1 day postoperatively and were recommended for discharge at 6 days postoperatively. The standard care protocol included the routine use of celecoxib 200 mg per day, as a painkiller, for 6 days postoperatively. 

### 2.3. Physical Therapy Programs

Both programs included 40 min outpatient sessions, once per week for 3 months postoperatively, under the supervision of a physiotherapist. The patients were also encouraged to perform home exercises as instructed by a physiotherapist in both programs, resulting in equivalent exercise duration and intensity for both programs. 

#### 2.3.1. Conventional Program

The conventional physical therapy program comprised stretching of the hamstrings, iliopsoas, quadriceps, and gluteus maximus; strength training of the core muscles and lower muscles; and education on sitting and standing postures and lifting movements. The contents of the outpatient sessions and a menu of home exercises were tailored for each patient by a physiotherapist.

#### 2.3.2. LBP Program

The LBP program comprised 20 min of the shortened conventional program and 20 min of a new program for improving LBP. In the latter, patients completed four types of exercises to improve spinal flexibility (lumbar flexion and extension, thoracic extension, spinal rotation in the lateral position, and spinal rotation in the tabletop position) and 3 min of aerobic exercises, including antero-posterior and lateral stepping exercises with careful monitoring of Borg’s scale and back pain [19,20,21]. In the home exercise portion of the LBP program, patients were instructed to perform two types of exercises to improve spinal flexibility (spine flexion and extension, and spinal rotation in the tabletop position) and 5–30 min of walking exercise. 

### 2.4. Clinical Evaluations

#### 2.4.1. Preoperative Data

From medical records, data regarding the age at surgery, sex, height, weight, body mass index (BMI), comorbidities (i.e., diabetes mellitus, hypertension, mental disorders, cardiac disorders, renal disorders, respiratory disorders, gastrointestinal disorders, hepatic disorders, cerebrovascular disorders, and history of malignant tumors), and diagnosis (LDH or LSS) were collected.

#### 2.4.2. Clinical Scores

The preoperative Japanese Orthopedic Association (JOA) score for degenerative lumbar disease, as a physician-assessed severity score, and the Oswestry Disability Index (ODI) and three domains of the EuroQoL-5 dimensions 5 levels (EQ-5D), as patient-reported scores, were collected preoperatively and at 3 months postoperatively [22,23,24]. Regarding the EQ-5D, the domains of mobility, pain/discomfort, and anxiety/depression were used in the current study as patient-oriented parameters of movement, pain, and mental-related QOL, respectively. The domain scores were converted into a weighted index according to a previous report [25,26].

#### 2.4.3. Satisfaction Scale

The overall treatment satisfaction was evaluated at 3 months postoperatively using Likert scale. All patients were asked, “Are you satisfied with the overall treatment?” on a self-completed questionnaire with a scale of 1–5: 1, satisfied; 2, slightly satisfied; 3, neutral; 4, slightly unsatisfied; and 5, unsatisfied. The treatment satisfaction ratings were binarized for analysis as follows: 1, satisfied; and 2–5, other.

### 2.5. Statistical Analysis

#### 2.5.1. Primary Analysis

The primary analysis evaluated the relationship between the physical therapy program and treatment satisfaction. All patients who failed to complete physical therapy more than once per week were excluded from analysis. First, univariate comparisons of age, sex, weight, height, BMI, comorbidities, diagnoses, and clinical scores between the control and test cohorts were performed using Student’s *t* test for continuous variables and the Chi-squared test or Fisher’s exact test for categorical variables. Subsequently, the outcomes, including clinical scores and treatment satisfaction, were compared between the two cohorts using the Student’s *t* test or Chi-squared test. Finally, multinomial logistic regression modeling was performed. A response of “satisfied” was set as a dependent variable, and age (≥60 or <60 years), sex (male or female), diagnosis (LSS or LDH), and physical therapy program (conventional or LBP) were included as explanatory variables. Adjusted odds ratios (aORs) and 95% confidence intervals (CIs) of dependent variables were calculated. 

#### 2.5.2. Secondary Analysis

The secondary analysis identified factors that were related to treatment satisfaction. Firstly, all patients in both cohorts were divided into two groups according to their treatment satisfaction; the satisfied group included patients with a satisfaction rating of 1, and the other group included patients with a satisfaction rating of 2–5. Postoperative clinical scores were compared between the satisfied and other groups. Finally, variables with a significance of *p* < 0.05 in the univariate analysis were included in the multivariate logistic regression model, with membership in the satisfied group set as the dependent variable.

#### 2.5.3. Analysis Settings

All analyses were performed using SPSS software (version 23; IBM Corp., Armonk, NY, USA). A *p*-value of <0.05 was considered statistically significant.

## 3. Results

### 3.1. Primary Analysis (Comparisons between the Two Cohorts)

#### 3.1.1. Patient Population 

Among the 100 consecutive patients initially enrolled into each cohort, eight and 16 patients in the control and test cohorts, respectively, were omitted because of inadequate compliance (Figure 2). Finally, the control cohort comprised 92 patients (mean age, 58.6 ± 19.0 years; 37 female patients) and the test cohort comprised 84 patients (mean age, 61.4 ± 17.1 years; 23 female patients). 

#### 3.1.2. Comparisons in Background Data

There were no significant differences between the cohorts in age, sex ratio, BMI, proportion of patients with each of the evaluated comorbidities, the number of surgical levels, and the proportion of each diagnosis (Table 1). In addition, the cohorts did not significantly differ in preoperative clinical scores, including the JOA score, EQ-5D mobility, pain, and anxiety scores, and the ODI.

#### 3.1.3. Comparisons in Clinical Outcomes

There were no significant differences in the postoperative JOA score, EQ-5D mobility and anxiety scores, and ODI between the two cohorts. However, the EQ-5D pain score was significantly higher (i.e., better) in the test than in the control cohort (−0.02 ± 0.02 vs. −0.03 ± 0.03, *p* = 0.029, Table 2). Furthermore, the number of patients who indicated that they were “satisfied” with the overall treatment was significantly higher in the test than in the control cohort (70.2% vs. 55.4%, *p* = 0.045). In the logistic regression analysis, age, sex, and physical therapy program were independent significant factors of treatment satisfaction (Table 3). Patients treated with the LBP program were, on average, more satisfied than those treated with the conventional physical therapy program (aOR = 2.34, *p* = 0.012).

### 3.2. Secondary Analysis (Factors Related to Overall Treatment Satisfaction)

There were 110 patients who indicated satisfaction with the treatment (satisfied group) and 66 who indicated otherwise (other group). In the univariate analysis, there were significant differences in the postoperative JOA and EQ-5D mobility, pain, and anxiety scores between the two groups (*p* = 0.014, 0.002, >0.001, and 0.010 respectively; Table 4). In the logistic regression analysis, the postoperative EQ-5D pain score was the only factor that was significantly related to treatment satisfaction (Table 5). Patients with worse postoperative pain scores were less satisfied than those with better postoperative pain scores (aOR = 3.09 × 1013, *p* = 0.002).

## 4. Discussion

In the current prospective clinical trial, we found that postoperative pain was associated with treatment satisfaction, independently of residual neurological symptoms and mental status, after minimally invasive decompression. Additionally, patients who were treated with a physical therapy program focused on LBP showed significantly lower pain and significantly higher treatment satisfaction than those who were treated with a conventional program.

There is a close relationship between postoperative pain management and treatment satisfaction [2,10,27]. According to a study by Levin et al., “staff were always doing everything they could to help with postoperative pain” was the strongest predictor of overall satisfaction in 453 patients who underwent lumbar spine surgery [27]. Additionally, Maher et al. analyzed the data of 562 patients who underwent spinal surgery and found an association between increased total perioperative opioid dosage and overall satisfaction [10]. Consistent with these previous studies, the secondary analysis in the current study showed that postoperative patient-reported pain was significantly correlated with overall treatment satisfaction. Thus, postoperative pain management and/or an effort to decrease the postoperative pain are critical for satisfaction, even after minimally invasive lumbar decompression. However, this may accelerate the use of postoperative pharmacological treatment, including opioids, which is currently a major social problem in the United States [28,29]. Hence, healthy alternative methods for pain management after minimally invasive lumbar decompression must be established. 

LBP is one of the world’s leading causes of function loss. Physical inactivity owing to LBP costs healthcare systems USD 54 billion worldwide, with productivity losses of USD 14 billion and 13.4 million disability-adjusted life-years [30,31]. For chronic LBP, rehabilitation therapy, including physical therapy, is the gold standard treatment for reducing pain and improving physical function [32,33]. In a randomized control study of 201 patients with chronic LBP, Shirado et al. found that physical therapy was more effective than pharmacological therapy in both reducing pain and gaining function [34]. In addition to these previous data, the current results suggest that our new physical therapy program, which is similar to previous programs for treating chronic LBP, is also effective for reducing postoperative LBP. Furthermore, the overall treatment satisfaction was significantly higher in patients treated with this type of program than in those treated with a conventional program. These results could be critical factors in establishing the postoperative management of patients treated with minimally invasive lumbar decompression. 

The relationship between LBP and psychological factors is well reported [35]. A previous study found a significant relationship between LBP and improvement in mental-related QOL after minimally invasive lumbar decompression [17]. Mental-related QOL has been reported as one of the most important factors in patient satisfaction [36]. Hence, we hypothesized that treating postoperative LBP might improve the mental-related QOL, thus resulting in further improvement in the overall treatment satisfaction. However, the current results failed to demonstrate this hypothesized relationship; it is possible that the anxiety/depression domains of the EQ-5D do not adequately reflect the mental-related QOL, and further studies using other patient-oriented parameters, such as the 36-item Short-Form Health Survey, are necessary [37].

There are several clinical implications relevant to spine surgeons. First, the current results suggest options for postoperative management that could reduce pain more effectively and achieve higher overall satisfaction after minimally invasive lumbar decompression, which is one of major surgeries for spine surgeons. Additionally, a relatively minor modification of the conventional physical therapy program, without extra medical cost, was enough to achieve higher satisfaction. Second, a previous high-evidence study showed that home-based exercise significantly improved chronic LBP. Hence, encouraging patients to perform a short course of physical exercise at home may also improve the postoperative LBP and satisfaction to some extent. Finally, although we did not evaluate the postoperative medicines used, the current intervention is potentially able to reduce the use of postoperative painkillers, including opioids, which is an urgent issue for spine surgeons.

Several limitations of the present study need to be addressed. First, the number of patients excluded from the final analysis was relatively high (8/100 in the control cohort and 16/100 in the test cohort) because we excluded patients who failed to complete physical therapy at least once per week. We could not obtain postoperative information, including satisfaction, from such patients. Additionally, the higher lost-to-follow-up rate in the latter cohort may have been affected by the COVID-19 pandemic [38]. Second, the overall satisfaction was only assessed at 3 months postoperatively. The current results must be validated in future studies with a long-term follow-up period. Third, the education of therapists was performed only before the LBP program, which might have resulted in bias because these therapists could gain new knowledge. Finally, we did not conduct a randomized controlled study, but a prospective comparative study. As the majority of studies on treatment satisfaction after spine surgery are retrospective analyses [2], the prospective nature of the current study could be considered as one of its strengths. However, the non-randomized design of the current study rendered it difficult to completely exclude the selection bias, which was the biggest limitation of the current trial. In addition, we did not calculate the sample power in advance.

## 5. Conclusions

Postoperative pain was correlated with treatment satisfaction, independently of residual neurological symptoms and mental status, after microendoscopic lumbar decompression surgery. Additionally, the current results clearly demonstrate that patients treated with a physical therapy program focused on LBP improvement showed significantly lower pain scores and significantly higher overall satisfaction than those treated with a conventional postoperative physical therapy program. We believe that these results could guide spine surgeons in achieving higher short-term satisfaction after microendoscopic lumbar decompression surgery, without an enormous extra medical cost.

## Figures and Tables

**Figure 1 jcm-11-05566-f001:**
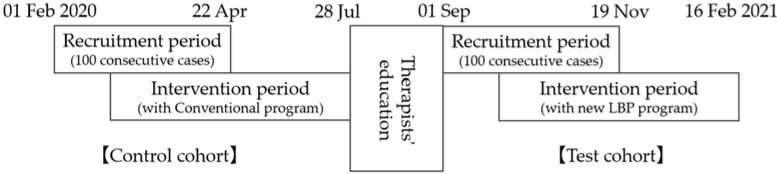
Time courses of the two prospectively enrolled cohorts.

**Figure 2 jcm-11-05566-f002:**
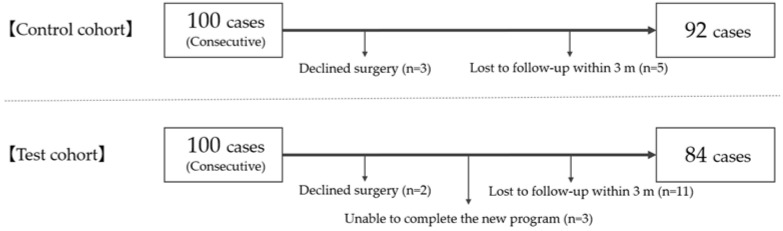
Patient population in each cohort.

**Table 1 jcm-11-05566-t001:** Univariate comparisons in background factors between the control and test cohorts.

	Control Cohort	Test Cohort	*p*-Value
Numbers	92	84	
Age (years)	58.6 ± 19.0	61.4 ± 17.2	0.310 ^#^
Sex (female/male)	37 (40.2%)/55 (59.8%)	23 (27.4%)/61 (72.6%)	0.082 ^†^
Height (cm)	162.0 ± 9.2	164.1 ± 9.9	0.155 ^#^
Weight (kg)	64.3 ± 11.3	65.1 ± 13.1	0.693 ^#^
BMI (kg/m^2^)	24.4 ± 3.3	24.0 ± 3.5	0.425 ^#^
Comorbidities (cases)			
Diabetes	14 (15.2%)	14 (16.7%)	1.000 ^†^
Hypertension	36 (39.1%)	26 (31.0%)	0.273 ^†^
Mental disorders	2 (2.2%)	1 (1.2%)	1.000 ^‡^
Cardiac disorders	8 (8.7%)	11 (13.1%)	0.467 ^‡^
Renal disorders	0 (0.0%)	3 (3.6%)	0.107 ^‡^
Respiratory disorders	1 (1.1%)	4 (4.8%)	0.194 ^‡^
Gastrointestinal disorders	4 (4.3%)	1 (1.2%)	0.370 ^‡^
Haptic disorders	3 (3.3%)	5 (6.0%)	0.481 ^‡^
Cerebrovascular disorders	7 (7.6%)	1 (1.2%)	0.066 ^‡^
Malignant tumor	2 (2.2%)	2 (2.4%)	1.000 ^‡^
Diagnosis (cases)			0.226 ^†^
Lumbar disc herniation	54 (58.7%)	41 (48.8%)	
Lumbar spinal stenosis	38 (41.3%)	43 (51.2%)	
Numbers of surgical level	1.2 ± 0.5	1.3 ± 0.6	
Preop clinical scores			
JOA score	13.7 ± 5.5	14.7 ± 4.9	0.210 ^#^
EQ-5D mobility	−0.14 ± 0.07	−0.12 ± 0.07	0.101 ^#^
EQ-5D pain	−0.11 ± 0.05	−0.10 ± 0.05	0.189 ^#^
EQ-5D anxiety	−0.07 ± 0.06	−0.06 ± 0.06	0.317 ^#^
ODI	47 ± 18	43 ± 17	0.127 ^#^

^#^: Student’s *t* test, ^†^: Chi-squared test, ^‡^: Fisher’s exact test. BMI: body mass index, JOA: Japanese Orthopedic Association, EQ-5D: EuroQoL-5 dimensions 5 levels, ODI: Oswestry Disability Index.

**Table 2 jcm-11-05566-t002:** Univariate comparisons in outcomes between the control and test cohorts.

	Control Cohort	Test Cohort	*p*-Value
Postop clinical scores			
JOA score	27.4 ± 1.9	27.2 ± 2.0	0.612 ^#^
EQ-5D mobility	−0.02 ± 0.05	−0.02 ± 0.03	0.297 ^#^
EQ-5D pain	−0.03 ± 0.03	−0.02 ± 0.02	0.029 ^#^
EQ-5D anxiety	−0.01 ± 0.02	−0.01 ± 0.03	0.235 ^#^
ODI (%)	12 ± 14	9 ± 10	0.216 ^#^
Overall treatment satisfaction			0.045 ^†^
Satisfied (1)	51 (55.4%)	59 (70.2%)	
Others (2–5)	41 (44.6%)	25 (29.7%)	

^#^: Student’s *t* test, ^†^: Chi-squared test. JOA: Japanese Orthopedic Association, EQ-5D: EuroQoL-5 dimensions 5 levels, ODI: Oswestry Disability Index.

**Table 3 jcm-11-05566-t003:** Logistic regression modeling of the factors related to satisfaction.

Objective Variable: Satisfied (Overall Treatment Satisfaction)				
Explanatory Variables	Reference	aOR	*p*-Value	95% CI
Age (≥60 years)	<60 years	0.31	0.003	0.14–0.68
Sex (Male)	Female	0.49	0.047	0.24–0.99
Diagnosis (LSS)	LDH	1.54	0.266	0.72–3.32
LBP program	Conventional program	2.34	0.012	1.20–4.55

LSS: lumbar spinal stenosis, LDH: lumbar disc herniation, aOR: adjusted odds ratio, CI: confidential interval.

**Table 4 jcm-11-05566-t004:** Univariate comparisons in scores between the satisfied and other groups.

	Satisfied Group (n = 110)	Other Group (n = 66)	*p*-Value
Postop clinical scores			
JOA score	27.6 ± 1.1	26.7 ± 2.2	0.014 ^#^
EQ-5D mobility	−0.01 ± 0.03	−0.04 ± 0.05	0.002 ^#^
EQ-5D pain	−0.02 ± 0.02	−0.04 ± 0.03	>0.001 ^#^
EQ-5D anxiety	−0.01 ± 0.02	−0.02 ± 0.03	0.010 ^#^
ODI	45 ± 18	45 ± 16	0.830 ^#^

^#^: Student’s *t* test. JOA: Japanese Orthopedic Association, EQ-5D: EuroQoL-5 dimensions 5 levels, ODI: Oswestry Disability Index.

**Table 5 jcm-11-05566-t005:** Logistic regression modeling of the scores related to satisfaction.

Objective Variable: Satisfied (Overall Treatment Satisfaction)			
Explanatory Variables	aOR	*p*-Value	95% CI
Postop JOA score	1.09	0.431	0.88–1.35
Postop EQ-5D mobility	3.02 × 10	0.599	0.00–9.92 × 10^6^
Postop EQ-5D pain	3.09 × 10^13^	0.002	1.38 × 10^5^–6.95 × 10^21^
Postop EQ-5D anxiety	1.80 × 10	0.784	0.00–6.00 × 10^7^

JOA: Japanese Orthopedic Association, EQ-5D: EuroQoL-5 dimensions 5 levels, ODI: Oswestry Disability Index, aOR: adjusted odds ratio, CI: confidential interval.

## Data Availability

The data presented in this study are available on request from the corresponding author.

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
