# Peer review of "Postoperative Physical Therapy Program Focused on Low Back Pain Can Improve Treatment Satisfaction after Minimally Invasive Lumbar Decompression"

_jcm, 2022, doi:10.3390/jcm11195566_

Round 1
Reviewer 1 Report
The paper is well written and easy-to-read. The argument is of interest. There are some major limitations that need to be solved.
Page 1 Keywords: Please include keywords that are mesh terms related to the paper content.
Page 2 line 67: This study is clearly not an observational trial (‘prospective cohort study'). You apply an intervention modification to the patients, they receive a different kind of physical therapy. This makes it an experimental trial without randomization. Actually, it is a Quasi-randomized controlled trial. You also know it because you state it in the limitation section (page 8 lines 290-294).
Page 3 lines 113-114: You perform 3 minutes of ‘aerobic exercises’, did you check the heart rate to see the effect? Did you monitor the perceived effort of the patient? Why only 3 minutes of aerobic exercise, from where comes this length?
Page 3 line 135: The satisfaction scale is actually a Likert scale [1-5], please use the correct name in the paragraph.
Page 4 Statistics paragraph: Did you calculate a priori sample size? If yes please report the calculation, if not it’s a limitation and you should report it.
Page 4 line 148: I don’t understand why you use Mann-Whitney for continuous variables, you should use a test for continuous variables like Student T! The Chi-Square test is a test for nominal variables, you have only ordinal variables where you can check differences using a Mann-Whitney test.
Page 2-4: I suppose this study needs ethical approval, did you receive it? Why do you not report it?
Author Response
Responses to Reviewer 1:
The paper is well written and easy-to-read. The argument is of interest. There are some major limitations that need to be solved.
Response: The authors would like to thank the reviewer for their constructive critique to improve the manuscript. We have made every effort to address the issues raised and to respond to all comments. The revisions are indicated highlighted in the revised manuscript. Please, find next a detailed, point-by-point response to the reviewer's comments. We hope that our revisions will meet the reviewer’s expectations.
Page 1 Keywords: Please include keywords that are mesh terms related to the paper content.
Response: We completely agree with this comment. Therefore, we have revised the keywords and added only MeSH terms. We would like to thank the reviewer for this insightful suggestion.
Page 2 line 67: This study is clearly not an observational trial (‘prospective cohort study'). You apply an intervention modification to the patients, they receive a different kind of physical therapy. This makes it an experimental trial without randomization. Actually, it is a Quasi-randomized controlled trial. You also know it because you state it in the limitation section (page 8 lines 290-294).
Response: We would like to thank the reviewer for this comment. We agree with the reviewer that our study is not an observational study. We understand that the term “cohort study” can also mean an intervention study, but we realize that it is not correct in the narrow sense. In contrast, the term “quasi-randomized controlled study,” which was suggested by the reviewer, might not have been adequate for us in the narrow sense, as we did not have any randomized procedures. Therefore, after consulting the professional epidemiologist, we have reworded our study design as follows:
“A prospective, non-randomized clinical trial was conducted (Figure 1). Initially, 100 consecutive patients undergoing microendoscopic decompression for lumbar disc herniation (LDH) or lumbar spinal stenosis (LSS) between February 1, 2020 and April 22, 2020 were enrolled into the control cohort.” (Lines 67–70)
We would like to thank the reviewer for giving us the chance to improve our manuscript. (Page 2, Line 68).
Page 3 lines 113-114: You perform 3 minutes of ‘aerobic exercises’, did you check the heart rate to see the effect? Did you monitor the perceived effort of the patient? Why only 3 minutes of aerobic exercise, from where comes this length?
Response: We would like to thank the reviewer for the questions. Please note that we monitored patients’ Borg’s scales, which include the heart rate and back pain carefully in all rehabilitation procedures. We have provided this information in the revised manuscript as follows:
“In the latter, patients completed four types of exercises to improve spinal flexibility (lumbar flexion and extension, thoracic extension, spinal rotation in the lateral position, and spinal rotation in the tabletop position) and 3 min of aerobic exercises, including antero-posterior and lateral stepping exercises with careful monitoring of Borg’s scale and back pain.” (Page 3, Lines 115–119)
Concerning the length of aerobic exercise, we decided the time based on a previous report [1].
Reference
- Shnayderman I, Katz-Leurer M. An aerobic walking programme versus muscle strengthening programme for chronic low back pain: a randomized controlled trial. Clin Rehabil. 2013;27(3):207-214. doi: 10.1177/0269215512453353
Page 3 line 135: The satisfaction scale is actually a Likert scale [1-5], please use the correct name in the paragraph.
Response: We agree with the reviewer’s comment. Please note that we have added the name of the scale on our revised manuscript as follows:
“The overall treatment satisfaction was evaluated at 3 months postoperatively using Likert scale.” (Page 4, Lines 143–144)
Page 4 Statistics paragraph: Did you calculate a priori sample size? If yes please report the calculation, if not it’s a limitation and you should report it.
Response: We would like to thank the reviewer for reminding us this important issue. We did not calculate the sample power in advance. Hence, we have discussed this issue in the revised manuscript as follows:
“However, the non-randomized design of the current study rendered it difficult to completely exclude the selection bias, which was the biggest limitation of the current trial. In addition, we did not calculate the sample power in advance.” (Page 8, Lines 304–306)
Page 4 line 148: I don’t understand why you use Mann-Whitney for continuous variables, you should use a test for continuous variables like Student T! The Chi-Square test is a test for nominal variables, you have only ordinal variables where you can check differences using a Mann-Whitney test.
Response: We would like to thank the reviewer pointing out our descriptive error. We have carefully performed statistical analysis in the current study and concluded that the correct name of our analysis was Student’s t test, as the reviewer indicated. We have revised our text accordingly.
“First, univariate comparisons of age, sex, weight, height, BMI, comorbidities, diagnoses, and clinical scores between the control and test cohorts were performed using Student’s t test for continuous variables and the Chi-squared test or Fisher’s exact test for categorical variables.” (Page 4, Lines 153-156).
Page 2-4: I suppose this study needs ethical approval, did you receive it? Why do you not report it?
Response: We obtained the IRB approval and wrote it to the indicated place according to the author’s guideline. We have provided this information as follows:
“The study was conducted in accordance with the Declaration of Helsinki and approved by the Institutional Review Board of Shimada hospital (No. 2020-012).” (Page 9, Lines 322–324)
Reviewer 2 Report
The authors examined specific low back pain (LBP) physiotherapy (PT) program compared to the conventional PT after microendoscopic lumbar decompression. The study is well designed and executed, statistical analyses are coherent and the manuscript is well written. I have only minor comments regarding the manuscript.
As the authors state, the study is not an RCT which is a limitation.
Abstract: more precise results than only p-values would be informative regarding the main results, for example regarding treatment satisfaction.
Introduction is well thought.
Methods: Could the education of physiotherapists affect the results of the study and better results of LBP program? Physiotherapists could get information regarding modern rehabilitation, for example. If the authors do consider this as a possible factor then I would suggest to state this in Discussion.
Conventional vs LBP program. The content of LBP program has been explained in detail, conventional has not. Did patients in LBP program more home exercises than conventional? How much exercises did patients in conventional program do compared to LBP program?
Why age has been dichotomized in the analyses?
Results: Table 1 could be more readable if percents were visible (sex, comorbidities, for example).
Discussion: There are some limitations in this study but the authors have already stated these. Higher lost-to-follow-up rate is truly one limitation. Would there be other reasons than coronavirus for this? Could it be that some patients are not able to execute more precise LBP program? If this is so, this would weaken the results. Some information regarding patient satisfaction or pain status would have been interesting from the patients that have dropped from the study.
Author Response
Responses to Reviewer 2:
The authors examined specific low back pain (LBP) physiotherapy (PT) program compared to the conventional PT after microendoscopic lumbar decompression. The study is well designed and executed, statistical analyses are coherent and the manuscript is well written. I have only minor comments regarding the manuscript.
Response: The authors would like to thank the reviewer for their constructive critique to improve the manuscript. We have made every effort to address the issues raised and to respond to all comments. The revisions are indicated highlighted in the revised manuscript. Please, find next a detailed, point-by-point response to the reviewer's comments. We hope that our revisions will meet the reviewer’s expectations.
As the authors state, the study is not an RCT which is a limitation.
Response: We agree with the reviewer that the study design was the biggest limitation of our study. We have reworded the limitation section to reinforce this issue as follows:
“However, the non-randomized design of the current study rendered it difficult to completely exclude the selection bias, which was the biggest limitation of the current trial. In addition, we did not calculate the sample power in advance.” (Page 8, Lines 304–306)
Abstract: more precise results than only p-values would be informative regarding the main results, for example regarding treatment satisfaction.
Response: We completely agree with the reviewer that we should have demonstrated the actual values of our primary outcomes in the Abstract. Therefore, we have provided this information accordingly.
“There were no significant differences in background factors; however, the patient-reported pain score at 3 months postoperatively was significantly better, and treatment satisfaction was significantly higher in the test than in the control cohort (-0.02±0.02 vs. -0.03±0.03, p=0.029; 70.2% vs. 55.4%, p=0.045, respectively).” (Page 1, Lines 20-23)
Introduction is well thought.
Response: We would like to thank the reviewer for the positive evaluation of our work.
Methods: Could the education of physiotherapists affect the results of the study and better results of LBP program? Physiotherapists could get information regarding modern rehabilitation, for example. If the authors do consider this as a possible factor then I would suggest to state this in Discussion.
Response: We never thought in this direction. However, this is a completely correct point. Therefore, we have added the statement regarding this issue in the limitation section as follows:
“Third, the education of therapists was performed only before the LBP program, which might have resulted in bias because these therapists could get new knowledge.” (Page 8, Lines 298–300)
Conventional vs LBP program. The content of LBP program has been explained in detail, conventional has not. Did patients in LBP program more home exercises than conventional? How much exercises did patients in conventional program do compared to LBP program?
Response: We would like to thank the reviewer for the questions. The conventional program consisted of stretching of the hip muscles (i.e., hamstrings, quadriceps, iliopsoas, gluteus maximus), trunk endurance training, and lower muscles strength training (e.g., squat). Both the LBP and the conventional program had durations of 40 min/session. Home exercises in the conventional program were performed for approximately 15 min every day. Therefore, exercise duration and intensity were equivalent for both programs. We have added the following part to the revised manuscript:
“Both programs included 40-min outpatient sessions, once per week for 3 months postoperatively, under the supervision of a physiotherapist. The patients were also encouraged to perform home exercises as instructed by a physiotherapist in both pro-grams, resulting in equivalent exercise duration and intensity for both programs.” (Page 3, Lines 101–104)
Why age has been dichotomized in the analyses?
Response: We would like to thank the reviewer for the question. We could use “age” as it is (continuous variables) in our multivariate logistic regression analysis. However, we thought that the results of the binarized variables would be easier to understand for the reader than those of continuous variables. Especially, our findings indicated that the patients aged <60 years tended to be satisfied to the treatment three times higher compared to those aged ≥60 years. We hope this reason would be accepted by the reviewer. However, we are willing to change to the result with “age” as the continuous variable if needed.
Results: Table 1 could be more readable if percents were visible (sex, comorbidities, for example).
Response: We agree with this suggestion. Accordingly, we have added the percentage value to Table 1. We would like to thank the reviewer for the insightful comment.
Discussion: There are some limitations in this study but the authors have already stated these. Higher lost-to-follow-up rate is truly one limitation. Would there be other reasons than coronavirus for this? Could it be that some patients are not able to execute more precise LBP program? If this is so, this would weaken the results. Some information regarding patient satisfaction or pain status would have been interesting from the patients that have dropped from the study.
Response: We would like to thank the reviewer for the insightful comment. As the reviewer pointed out, we agree that the short follow-up period was one of our biggest limitations in the current study. This was attributed to a restriction of the Japanese insurance system. Namely, the Japanese insurance system allows patients to have physical therapy without any limitation within 3 months postoperatively. However, after this period, the selected patients (those with severe symptoms) were only able to continue the physical therapy. Therefore, we decided to put the outcome period as 3 months after surgery to reduce the bias. Regarding the information of whom dropped out from current study, we do not have enough data, including satisfaction data, because we cannot obtain them from patients. Therefore, we cannot demonstrate the data on our manuscript. We have revised the corresponding part in the Discussion section as follows:
“We could not obtain postoperative information, including satisfaction, from such patients.” (Page 7, Lines 294–295)
Round 2
Reviewer 1 Report
Why does the pValues of the table 4 did not change according to the change of the test you used?
Author Response
Responses to Reviewer 1:
Why does the pValues of the table 4 did not change according to the change of the test you used?
Response: Thanks for your question. We did not change the p-value on current revised manuscript. This is because it turns out that we used the “Student’s t test” rather than “Mann-Whitney U test” from our first analysis. We realized that the descriptive of “Mann-Whitney U test” was incorrect one. Hence, we changed the words of “Mann-Whitney U test” to “Student’s t test” without any change of p-value. We hope this would be convincing answer for this reviewer. Thanks for giving us a chance to correct our manuscript.